# Effects of Nutrition Counselling and Unconditional Cash Transfer on Child Growth and Family Food Security in Internally Displaced Person Camps in Somalia—A Quasi-Experimental Study

**DOI:** 10.3390/ijerph192013441

**Published:** 2022-10-18

**Authors:** Mohamed Kalid Ali, Renée Flacking, Munshi Sulaiman, Fatumo Osman

**Affiliations:** 1School of Health and Welfare, Dalarna University, 79182 Falun, Sweden; 2Food and Agriculture Organisation of the United Nations (FAO), Somalia Country Office, Nairobi P.O. Box 30470-00100, Kenya; 3BRAC International, Clock Tower, Kampala P.O. Box 31817, Uganda

**Keywords:** counselling, cash transfer, humanitarian setting, IDP, nutrition, underweight, wasting

## Abstract

The effects of nutrition counselling (NC) and unconditional cash transfer (UCT) in improving growth in children under five and household food security are poorly understood in humanitarian settings. Therefore, this study aimed to evaluate the effects of NC and NC combined with unconditional cash transfer (NC + UCT) on children’s growth and food security in Somalia. The study was performed with a quasi-experimental design in two districts in the Banadir region of Somalia. Caregivers (*n* = 255) with mildly to moderately malnourished children aged 6 to 59 months old (*n* = 184) were randomized to the NC, NC + UCT and control groups. The interventions consisted of weekly NC for three months alone or in combination with UCT. The outcome variables were wasting, underweight, stunting, and food security. Difference-indifferences analysis was used to estimate the effect of the interventions. Our study did not find any significant impacts of NC or NC + UCT on child wasting, underweight, stunting, food security or household expenses. In conclusion, NC, alone or in combination with UCT, did not impact children’s growth or household food security. Thus, a culturally tailored NC programme over a longer period, supplemented with cash transfer, could be beneficial to consider when designing interventions to reduce malnutrition and food insecurity.

## 1. Introduction

Malnourished children in Sub-Saharan Africa account for one-third of all undernourished children globally; approximately 39% of children are stunted, 10% wasted, and 25% underweight [1,2,3]. Malnutrition is a major risk factor in children under five and is associated with a higher disease burden and increased levels of mortality [4]. The rate of malnutrition across Somalia is alarming and remains among the worst in the world. Approximately 1.2 million children under five are acutely malnourished in Somalia, including 232,000 who suffer from severe acute malnutrition. Large population displacement happened in southern Somalia in the first half of 2017 as a result of drought, insecurity, and other factors. Data from United Nations High Commissioner for Refugees (UNHCR) indicated that an estimated 818, 000 people were displaced between January and June 2017, including 662,000 people (81% of the total) who were displaced due to drought and drought-related factors. Children are the ones who are affected the worst and they are at high risks for experiencing malnutrition, measles, malaria, Acute Respiratory Infections (ARI) and Acute Watery Diarrhoea (AWD) [5]. Furthermore, food insecurity is also a major concern for many developing countries. Households are defined as food secure when they have “access to sufficient, safe, nutritious food to maintain a healthy and active life” at all times [6].

Malnutrition has been shown to be multifactorial, and no single programme implemented in isolation is likely to be sufficient to sustain a significant reduction in undernutrition [7]. Nutrition counselling has often been part of strategies to reduce malnutrition in children and is mainly aimed at promoting and/or strengthening caregivers’ knowledge and practices regarding children’s health and nutrition. The existing nutrition education counselling program in Somalia for preventing malnutrition involves training community workers, who then counsel caregivers regarding the feeding of children [8]. Another frequently used intervention to address malnutrition in a humanitarian setting is cash transfer. Usually, cash transfers (CT) are direct payments, often from governments or humanitarian agencies, made to eligible groups of people. Cash transfers are grouped into three main categories: unconditional cash transfer (UCT), which is a cash transfer made without any conditions that the recipient must meet; conditional cash transfer, which is made on the condition that the recipient meets specified criteria such as school attendance or receiving vaccinations; and labelled cash transfer, in which funds are indicated, or “labelled” for specific purposes, but the conditions are not enforced [9].

Despite the acknowledged potential and the promising positive impacts of cash transfer programs, numerous reviews in the last decade indicate limited impacts of CT on child nutritional outcomes [10,11,12,13,14]. However, a recent meta-analysis evaluated 15 CT programs on their effect on children’s nutritional status. The findings showed that cash transfer targeted to households with young children improved children’s growth and contributed to reduced stunting [15]. Due to conflicting evidence, it is crucial to understand factors leading to successful or lack of CT impact. For example, Owusu-Addo et al. [13] hypothesize that factors such as water and sanitation, the size of the transfer, and household size might contribute to the lack of its effects. Furthermore, evidence has also shown that the pathway impacts of CT on child nutrition is yet to be clearly understood [16].

Cash transfer programs have become more common in Somalia over the past decade with the influx of local displacements, notably in the southern and central regions of the country. Most of the programs have been short-term projects designed principally to address the basic needs in humanitarian settings and are purely donor-led, targeting mainly IDPs. No previous studies have evaluated the impact of the provision of NC + UCT in Somalia that addresses simultaneously caregiver’s nutritional knowledge while supporting them with CT. Thus, this study aimed to evaluate the effects of NC and UCT on children’s growth and families’ food security in IDP camps in Somalia.

## 2. Materials and Methods

### 2.1. Study Design

This study had a quasi-experimental design with two intervention arms, NC with or without UCT, and one control arm. The effectiveness was assessed by measuring children’s growth and families’ food security pre- (baseline, July 2017) and post-intervention (follow-up, February 2019). Caregivers who had the responsibility for one or more children aged 6–59 months were the target group. The caregivers were mainly mothers but also included grandmothers or female guardians. The inclusion criteria were as follows: children aged six months up to five years who had mild or moderate malnutrition (mild, −1.0 to −1.9; moderate, −2.0 to 2.9 WHZ). Children with severe acute malnutrition (WHZ < −3) and apparent health problems that affected their nutrition such as measles, malaria, ARI, and AWD were excluded and referred to the nearest health facility for further screening and treatment. Participants were provided detailed explanations about the study objectives and issues related to data collection, and written consent was obtained before commencing the study. The study received ethical approval from the Research and Ethics Review Committee of the Federal Ministry of Health (approval no. MoH & HS/DGO/0129).

### 2.2. Setting

The study was conducted in six IDP camps in two districts, Dayniile and Dharkanley, that host the majority of IDPs in the Banadir region of Somalia. These districts were intentionally selected because they were areas with intervention programmes supported by an international NGO. Those who live in these IDP camps are the most vulnerable groups and are often subject to arbitrary evictions by the government, as well as to evictions by landowners. IDP housing is unregulated and has no formal status that is monitored by the government. Instead, they are managed by local managers known as “gatekeepers”. These gatekeepers act as brokers between IDPs, local authorities and nongovernmental organizations (NGOs). Residents in these IDP camps usually receive humanitarian assistance that are aimed to prevent deterioration of health, nutrition and food security situations.

### 2.3. Sample Size

Our sample size was determined based on a balance between operational feasibility under security, budget constraints and the goal for improving child nutrition status. The primary outcome for this study was the change in WHZ for individual children with WHZ less than two standard deviations below the median measurement for the reference group and is regarded as wasted or acutely undernourished [17]. There was no specific calculation of sample size for all the outcomes of interest for this paper. However, based on the original objectives of the study [8], the sample size was 1655 households that allows for the precision of WHZ in target districts assuming mean household size of 6.5, attrition rate of 15%, design effect of 2.0 and an alpha of 0.1 (90% CI) [18,19,20]. This was necessary to have sufficient power to demonstrate a minimum detectable effect size of 0.18 WHZ when comparing with any of the two interventions with the control group. However, due to unavailability of cash transfer to cover the beneficiaries in Baidoa district, we dropped the households of this district and only sub-samples of Daynile and Dharkanley districts are analysed in this paper.

### 2.4. Interventions

The intervention consisted of NC alone or in combination with UCT (NC + UCT) on infant and young child feeding (IYCF) was provided to both intervention groups for three months (August–October 2017). The NC sessions were based on the UNICEF/WHO IYCF guidelines [21], and the topics covered are listed in Table 1. Participants received NC during one-to-two-hour weekly NC sessions for 12 weeks. Each counselling session was attended by 20–22 caregivers. Eight trained community nutrition volunteers (CNVs) provided all the sessions. The CNVs were trained volunteers that conducted routine nutrition counselling and assessments and were supported by Save the Children and the Federal Ministry of Health. A small monthly stipend was provided to the CNVs to use for transportation and calls to caregivers (e.g., when inviting caregivers to the counselling sessions). The counselling sessions were conducted at health facilities. Before every session, caregivers were asked to share their experiences and strategies that helped them to improve their household nutrition practices. UCTs were provided, in addition to the NC, to one of the two intervention groups. The UCT involved a cash transfer of $40 per month for 3 months through mobile phones with registered Simcards provided by Save the Children. The cash transfers were provided to parents within one month after the NC intervention had been completed. Once follow-up survey completed, caregivers in the control groups have received nutrition counselling delivered by CNVs.

### 2.5. Data Collection

The baseline survey was collected in July 2017 and included 709 caregivers with 2836 children. Twelve weeks after the NC intervention was completed and 11 months after the last cash transfer in the NC + UCT group, follow-up data was collected from 255 caregivers of 184 children (Figure 1).

Data were collected by nine data collectors trained by Save the Children. They were all professional data collectors with several years of experience in data gathering for periodic monitoring and research projects. They received two days of refresher training from the first author in Mogadishu, Somalia, before starting data collection for both the baseline and follow-up surveys. The training focused on administering the study questionnaire, mobile data collection and how to conduct naturalistic observation. The data collectors conducted interviews at participants’ residences. Socioeconomic status was calculated based on a composite measure of household assets and amenities, such as the number of bedrooms, construction materials of the main house, ownership of household durable goods, type of household toilet used, and source of drinking water.

### 2.6. Primary and Secondary Outcomes Measures

The primary outcome variable was WHZ (wasting) in children under five. Child weight was measured to the nearest 100 g using a digital scale (Seca, Hamburg, Germany). The length of children <24 months of age was measured to the nearest 0.1 cm with a length board, and the height of children ≥24 months was measured to the nearest 0.1 cm with a portable stadiometer. We calculated z scores using the WHO child growth charts [22].

The secondary outcomes were the WAZ (underweight), the HAZ (stunting), and household food security indicators such as the household and child dietary diversity scores, household food consumption score, household hunger scale, minimum child dietary diversity score and household expenses. Food security status was measured from a set of questions that captured the domains of food access described in the Radimer framework [23]. To determine household food security, dietary diversity data were collected based on 24 h recall. This indicator is based on eight food groups with 13 categories of food. The categories were cereals; roots and tubers; vegetables; fruits; meat, poultry and offal; eggs; fish and seafood; pulses/legumes/nuts; milk and milk products; oil/fat; sugar/honey; and miscellaneous [24]. Each group of foods had a binary outcome equal to one if a household member consumed the food during the 24 h preceding the survey and zero otherwise. The score was calculated by summing the scores across the eight food groups, and it ranged from zero to eight.

### 2.7. Randomization

To reduce heterogeneity, we excluded IDP camps that had previously received similar interventions. We identified the targeted IDP camps based on a nutrition intervention programme that was implemented by Save the Children. Eligible caregivers and their children were randomized to one of the three arms with a uniform treatment allocation ratio: NC, NC + UCT and control. Stratification was performed to ensure an equal number of caregivers was assigned to each CNV delivering NC. STATA software was used to generate the random allocation sequence. Randomization was done at Save the Children’s office, and the procedures and group assignments were blinded to the enumerators.

### 2.8. Statistical Analysis

Both descriptive and inferential statistics were used for analysis. Frequencies and percentages were computed to describe the sociodemographic characteristics of the study sample.

Principal component analysis (PCA) was used to determine the socioeconomic status of respondents using the sociodemographic data on the households. PCA reduced the multidimensional nature of the data to a single score that was categorized into three groups: low, middle and upper [25]. An intention-to-treat (ITT) analysis was performed regardless of NC session attendance. Difference-in-differences (DiD) estimates were run using regression models to calculate differences in outcomes between the intervention and control groups for each indicator at baseline and follow-up. To obtain the WHZ, WAZ and HAZ, the ZSCORE06 command in STATA was used. In addition to the Z scores, we generated a composite index of anthropometric failure (CIAF), which was coded as 1 if a child was wasted, stunted, underweight or any combination of these and 0 if they were not wasted, stunted or underweight [26]. The CIAF provided a more global measure of undernutrition. To measure food security indicators such as the dietary diversity score, food consumption score, and household hunger scale, a methodology recommended by the Emergency Food Security Assessment Handbook was used [27]. To measure the effects of the interventions, we included an intervention interaction term with a follow-up dummy in the model. Significance was defined as *p* < 0.05. Data were analysed using STATA version 15 (Stata/SE 15.1 Stata Corporation, College Station, TX, USA).

## 3. Results

A total of 255 caregivers with 184 children were analysed. The baseline demographic characteristics of the population are presented in Table 2. There were no significant differences in wasting, underweight and stunting or among other food security indicators observed between the NC, NC + UCT and control groups. Out of the 12 sessions of NC, 66% (*n* = 85) of the caregivers attended all sessions, 26% (*n* = 33) attended 9–10 sessions, and information was lacking for 9 caregivers. All caregivers in the NC + UCT group received their cash transfer (Figure 1).

### 3.1. Growth Outcomes in Children

In the NC intervention group, we did not find any significant impact on wasting (aOR: 0.81, 95% CI: 0.26–2.56, *p*  =  0.722), underweight (aOR: 1.29, 95% CI: 0.40–3.48, *p*  =  0.603), or stunting (aOR: 0.84, 95% CI: 0.29–2.46, *p*  =  0.764) compared to the control group. Similarly, in the NC + UCT group, we did not find any significant reduction in wasting (aOR: 0.79, 95% CI: 0.26–2.36, *p*  =  0.67), underweight (aOR: 0.88, 95% CI: 0.33–2.35, *p*  =  0.81), or stunting (aOR: 1.51, 95% CI: 0.55–4.20, *p*  =  0.43) compared to the control group. Using a CIAF (being wasted, stunted, underweight or any combination of these), this study found no significant change in either NC (aOR: 1.21, 95% CI: 0.45–3.25, *p*  =  0.70) or NC + UCT arm (aOR: 1.1, 95% CI: 0.39–2.87, *p*  =  0.91). Similarly, no intervention effect was observed on the child dietary diversity score, minimum child dietary diversity (≥4 food groups) or household expenses (Table 3).

Additional sensitivity analyses on WHZ, WAZ and HAZ to check for the presence of intervention effects on anthropometry were conducted. The results of these analyses are presented in Appendix A, and no intervention effect was observed for any of the three anthropometric z scores.

### 3.2. Food Security and Household Expenses

The data in Table 4 present the food security characteristics of the intervention and control groups. At baseline, caregivers in all three groups reported consumption of at least 7 food groups in the past 24 h. There were no significant differences between baseline and follow-up in terms of dietary diversity score, household hunger scale score or household food consumption score in any of the groups (Table 4). Caregivers were also asked about expenses related to household food, water, fuel for food cooking, health and education, and no group showed a significant difference between baseline and follow-up.

## 4. Discussion

This is the first study to evaluate the impact of NC alone or in combination with UCT on caregivers of children with mild and moderate malnutrition. The findings from this study revealed that there were no statistically significant improvements in nutrition indicators such as wasting, stunting and underweight, measured in terms of WHZ, HAZ and WAZ, in the NC or NC + UCT group compared to the control group. Similarly, we did not detect statistically significant improvements in children’s dietary diversity scores or other household food security indicators, such as household dietary diversity, household hunger scale score, household food consumption score and household expenses for children’s health, food, water and education. The overall lack of statistically significant improvements in growth and food security indicators is likely due to the poor socioeconomic and feeding practices among the target beneficiaries. Moreover, NC that is not individualized and adapted to each mother-child pair after a thorough analysis of the child’s situation might not render positive results. The fact that our counselling was at the group level and not personalized and adapted to the socioeconomic situation of each woman likely contributed to the negative results [28].

Our study sought to understand the impact of NC with and without UCT on child growth. The lack of a significant reduction in wasting and underweight in our study was an unexpected finding but could be explained by the high mobility of IDPs, limited social services and high rate of loss to follow-up [29]. Hoddinott et al. [30] evaluated an implemented intervention consisting of CT coupled with nutrition behaviour change communication, which resulted in a positive impact on children’s nutritional outcomes, specifically on HAZ. However, the positive impact on children’s growth could be due to the nature of the study, which targeted farming communities where children had access to diversified food. Their intervention period also lasted two years, which was far longer than on our study [30]. CT programs implemented in some African countries have shown no significant improvements in children’s undernutrition status [31]. However, CT programs in Zambia, Malawi, Zimbabwe, Ethiopia and Kenya had improved household food security, which is an immediate determinant of malnutrition.

The link between UCT and nutrition is well researched and documented in several studies [32,33,34]. However, the amount of CT differs across countries. The limited significant impact we observed in our study could be related to the relatively low transfer size, given the evidence that a transfer level of approximately 20 per cent or more has shown to be associated with broader impacts across multiple social and productive domains [35]. On the other hand, it has been shown that when participants have been offered a large amount of money for a longer period, no improvements were detected in children’s nutritional status [36].

In a systematic review conducted by the Overseas Development Institute, the impact of cash transfer on child nutrition was examined [37]. Out of the five studies addressing the effect of cash transfer on child wasting, only one instance of conditional cash transfer was found to result in a statistically significant reduction in wasting (13%) among children aged 12–24 months. Furthermore, understanding intra-household allocation of resources is crucial specially on the way a CT programme impacts individuals within [38]. Evidence has shown that when households face an exogenous increase in household disposable income, adult members of the household may be prioritized compared to children since they are the ones who generate income for the household, or households resources are under their control [10,39]. Therefore, more research is needed to shed more light on this issue.

Somalia had no government led social protection scheme or system until recently and most of the CT are led by development actors and are subjected to the cash availability as a response to humanitarian crisis caused by recurrent climatic shocks. In the case of Somalia, the current Social SafetyNet scheme “Baxnaano” is only $20 USD, however, its effectiveness has not been documented so far.

Systematic reviews of NC topics such as breastfeeding promotion and complementary feeding education have suggested small to moderate effects on children’s nutritional status depending on the background level of food security [40,41]. However, these results confirm that interventions to improve IYCF have only modest effects on children’s nutritional status if not supplemented with additional food, micronutrients and other interventions that can improve children’s diets and women’s health and nutrition in the prenatal period [42]. Additionally, data from other impact evaluation studies found that wasting and stunting cases are attributable to foetal growth restrictions, and they have highlighted the need to implement interventions before and during pregnancy [43,44,45,46].

Another possible explanation for a lack of significant improvements on children’s growth is that the pre- and post-data collection periods occurred during dry seasons. Somalia endured an extreme drought, which resulted in famine and mass displacement in several regions of the country. Furthermore, changes in seasonal harvests always affect the food security situation [47]. In most humanitarian contexts, nutritional levels and food security conditions are generally worsened by little rain and extended drought over multiple seasons. Thus, during periods of extreme drought, the resources are so scarce that an NC, with or without UCT, may not be as beneficial as when the resources of food are ‘good-enough’. In future studies, attention should be paid to assessing locally available foods while maximizing household income to obtain the improved outcomes on children’s growth and households’ food security.

Based on these pieces of evidence from other studies, future integrated interventions in humanitarian settings should assess the quality of training and supervision offered to caregivers, the amount of cash transfer, the mobility of IDPs, and the health and nutrition services offered to caregivers and children.

### Strength and Limitations of the Study

The strength of this study is that, to our knowledge, this is the first study to assess the impact of NC in combination with cash transfer on child nutrition in a population exposed to displacement. This study has also considerable limitations. First, the study lacked power and was not adequate to assess the nutritional impact or to study the relationship between changes in nutrition and food security. This was due to dropping the sample of Baidoa district due to unavailability of project that can be leveraged with cash transfer. Second, the study was conducted in two distinct seasons, in which the intake of nutrients, WHZ and WAZ differed significantly, which adds to the external validity of the overall analysis. Third, half of the cash transfer recipients received cash transfer and NC late after randomization, and the data of these individuals was removed from the analyses. This reduced the original sample size and may have affected the study power. Fourth, overreporting of desirable dietary intake is a common occurrence in the assessment of feeding practices using recall techniques [48]. Fifth, the quality of the training and supervision of CNVs is unclear, which affects the fidelity of the intervention.

## 5. Conclusions

Our results revealed that NC, alone or in combination with UCT, did not significantly improve children’s growth or household food security. To improve the impact of NC, understanding the current existing capacity and community culture while providing NC during humanitarian emergencies not only helps us to identify current practices but also provides a foundation for making tailored plans to address any identified gaps. Careful consideration should be given to the amount of money transferred, acknowledging household sizes and constitutions. Tailoring future interventions to deliver culturally relevant NC, coupled with cash transfer, could improve nutritional outcomes. Future research should focus on tailoring culturally adaptive and context-specific linkages between cash transfer programmes and additional social services beyond nutrition counselling. Furthermore, understanding the intra-household allocation of resources and the consequent impacts for young children need to be researched.

## Figures and Tables

**Figure 1 ijerph-19-13441-f001:**
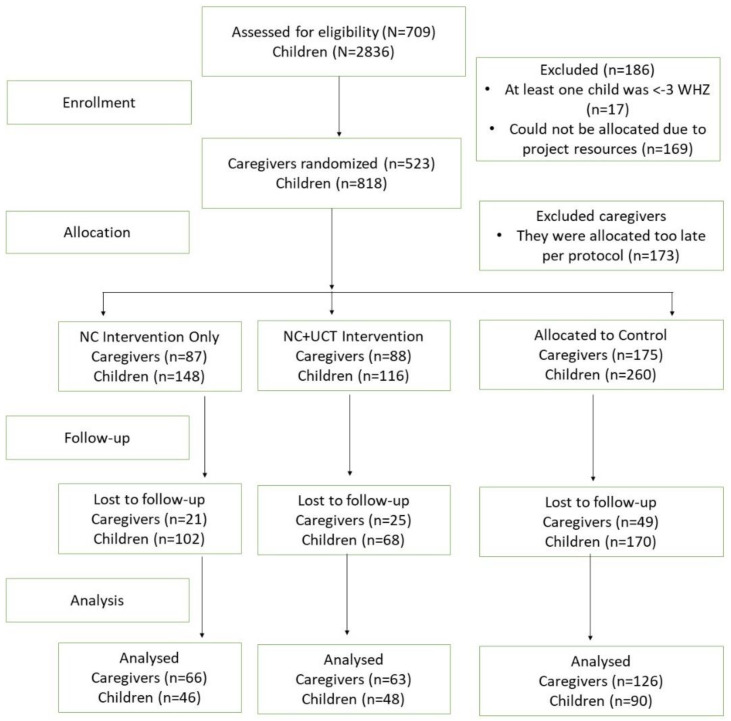
Flowchart.

**Table 1 ijerph-19-13441-t001:** Training topics covered in 12 Sessions.

Topics Covered
Early initiation Exclusive breastfeeding Breastfeeding and benefits Introduction of complementary feeding Continuous breastfeeding Dietary diversity and frequency Frequency of feeding for children aged 6–8 months Frequency of feeding for children aged 9–23 months Refusal to eat Sleep at night Diarrhoea/illness Treatment of water Handwashing Water and faeces management Bottle feeding and risks Vitamin A/deworming/full immunization

**Table 2 ijerph-19-13441-t002:** Baseline characteristics of the study arms.

Caregiver Variables		Baseline		
		NC (*n* = 87)	NC + UCT (*n* = 88)	Control (*n* = 175)
Districts	Dayniile, *n* (%)	22 (25.3)	22 (25)	44 (25.1)
	Dharkanley, *n* (%)	65 (74.7)	66 (75.0)	131 (74.9)
Caregiver age, *n* (%)	15–19	3 (3.5)	6 (6.8)	16 (9.1)
	20–29	37 (42.5)	64 (72.7)	89 (50.9)
	30–39	41 (47.1)	16 (18.2)	60 (34.3)
	40≥	6 (6.9)	2 (2.3)	10 (5.7)
Married caregivers, *n* (%)		84 (97.0)	86 (97.7)	155 (88.6)
Age at first marriage, (mean ± sd)		16.0 ± 1.8	16 ± 2.4	16.0 ± 2.2
Age at first childbirth, (mean ± sd)		17.0 ± 1.8	18.0 ± 2.4	17.0 ± 2.1
Socioeconomic status of household, *n* (%)	Low	28 (32)	26 (30)	42 (24)
	Middle	25 (28.7)	22 (25)	45 (25.7)
	Upper	34 (39.1)	40 (45.5)	88 (50.3)
Average number of children, (mean ± sd)		7 ± 3.5	5 ± 3.1	5 ± 3.1
Caregivers attended ANC during the last pregnancy ^a^, *n* (%)		27 (79.4)	32 (94.1)	56 (93.3)
Month visited at ANC during the first pregnancy ^a^, (mean ± sd)		5 ± 1.6	4 ± 1.5	4 ± 1.5
**Child Variables**		**NC** **(*n* = 148)**	**NC + UCT** **(*n* = 116)**	**Control** **(*n* = 260)**
Sex of child is female, *n* (%)		24 (52.2)	25 (52.1)	50 (55.6)
Age of the child in months, (mean ± sd)		34.7 ± 15.9	27 ± 15.3	29.7 ± 14.9

^a^ Number represents those who attended ANC.

**Table 3 ijerph-19-13441-t003:** Intervention effects on child growth and food security outcomes between baseline and follow-up.

Variables of Interest	NC		NC + UCT		Control		Effect of NC vs. Control	*p*	Effects of NC + UCT vs. Control	*p*
	Baseline	Follow-Up	Baseline	Follow-Up	Baseline	Follow-Up	Adjusted OR (95% CI) ^a^		Adjusted OR (95% CI) ^a^	
	*n* = 148	*n* = 46	*n* = 116	*n* = 48	*n* = 260	*n* = 90				
Wasting, *n* (%)	18 (12.2)	7 (15.2)	18 (15.5)	9 (18.7)	31 (11.9)	16 (17.8)	0.81 (0.26–2.56)	0.722	0.79 (0.26–2.36)	0.67
Underweight, *n* (%)	44 (29.7)	21 (45.7)	42 (36.2)	22 (45.8)	91 (35.00)	43 (47.8)	1.29 (0.40–3.48)	0.603	0.88 (0.33–2.35)	0.81
Stunting, *n* (%)	25 (16.9)	10 (21.7)	20 (17.2)	16 (33.3)	44 (16.9)	22 (24.4)	0.84 (0.29–2.46)	0.764	1.51 (0.55–4.20)	0.43
Composite index of anthropometric failure ^b^, *n* (%)	62 (41.9)	28 (60.8)	58 (50.0)	32 (66.7)	124 (47.7)	57 (63.3)	1.21 (0.45–3.25)	0.701	1.06 (0.39–2.87)	0.91
Children’s minimum dietary diversity (≥4 food groups), *n* (%)	18 (12.2)	28 (60.9)	9 (7.8)	22 (45.8)	31 (11.9)	52 (57.8)	1.16 (0.39–3.48)	0.779	1.01 (0.31–3.27)	0.99
Child dietary diversity score (mean ± sd)	3.3 ± 1.2	4.8 ± 1.6	2.9 ± 1.3	4.6 ± 2.0	3.2 ± 1.4	4.8 ± 1.9	−0.18 (−0.90)	0.612	0.03 (−0.78–0.84)	0.95

^a^ Differences in wasting, underweight and stunting at follow-up, adjusted for baseline differences in anthropometry. ^b^ The composite index of anthropometric failure includes six categories of undernutrition: wasting only, wasting and underweight, wasting, stunting and underweight, stunting and underweight, stunting only, and underweight only. A child is considered to have anthropometric failure if they are either wasted, underweight or stunted or any combination of these.

**Table 4 ijerph-19-13441-t004:** Intervention effects on the caregiver report-based dietary diversity score, household hunger scale score, household food consumption score and expenditure situation.

Indicator of Interest	NC		NC + UCT		Control		Effect of NC vs. Control	*p*	Effect of NC + UCT vs. Control	*p*
	Baseline *n* = 87	Follow-Up *n* = 66	Baseline *n* = 88	Follow-Up *n* = 63	Baseline *n* = 175	Follow-Up *n* = 126	Adjusted Coefficient (95% CI)		Adjusted Coefficient (95% CI)	
	Mean (95% CI)	Mean (95% CI)	Mean (95% CI)	Mean (95% CI)	Mean (95% CI)	Mean (95% CI)				
Household dietary diversity score (12 food groups)	6.9 (6.6–7.3)	7.2 (6.7–7.7)	6.8 (6.5–7.2)	7.4 (7.1–7.8)	6.8 (6.7–7.0)	7.4 (7.1–7.8)	−0.3 (−1.2–0.4)	0.31	−0.1 (−0.8–0.7)	0.90
Household hunger scale score	3.3 (3.1–3.6)	2.1 (1.7–2.5)	3.3 (3.1–3.6)	2.6 (2.2–3.0)	3.5 (3.3–3.7)	2.5 (2.2–2.8)	−0.2 (−0.9–0.4)	0.37	0.2 (−0.3–0.8)	0.55
Household food consumption score	10.5 (9.7–11.4)	10.9 (9.8–12.0)	10.0 (9.4–10.7)	11.8 (10.9–12.7)	10.8 (10.3–11.3)	11.8 (11.3–15.5)	−0.7 (−2.2–0.8)	0.35	0.7 (−0.7–2.0)	0.31
Household Expenses ^a^										
Food	46 (40.5–51.0)	46 (40.3–52.6)	41 (35.8–46.5)	43 (39.2–46.9)	43 (39.4–47.3)	39 (36.7–41.9)	4.7 (−3.7–13.1)	0.27	5.9 (−1.9–13.7)	0.14
Water	9 (6.8–10.3)	6 (4.8–6.6)	8 (6.2–9.2)	6 (4.5–7.2)	8 (7.2–9.6)	5 (4.3–5.8)	0.4 (−1.7–2.6)	0.69	1.4 (−0.7–3.6)	0.18
Fuel for cooking	9 (7.4–10.8)	9 (7.0–10.3)	7 (5.8–8.4)	7 (6.4–8.4)	8 (6.9–8.8)	7 (6.5–7.9)	0.21 (−2.3–2.8)	0.86	0.94 (−0.9–2.8)	0.34
Education	4 (2.2–5.2)	5 (3.1–6.9)	2 (0.5–2.8)	2 (0.7–2.9)	3 (1.5–3.9)	4 (2.7–5.3)	−0.1 (−3.2–3.1)	0.96	−1.2 (−3.7–1.3)	0.34
Health expenses for children under 18	4 (2.5–5.7)	4 (1.7–7.3)	3 (1.4–3.6)	1 (0.4–2.2)	2 (1.4–3.0)	2 (0.8–2.3)	0.3 (−3.9–4.4)	0.90	−0.7 (−2.6–1.1)	0.41
Health expenses for all members over 18	4 (2.3–5.6)	2 (0.9–3.6)	3 (1.8–4.5)	1 (0.03–1.3)	3 (1.7–4.7)	2 (0.7–2.2)	−0.0 (−2.6–2.5)	0.98	−0.8 (−3.0–1.4)	0.47

^a^ Amount in USD.

## Data Availability

The raw data presented in this study are available on request from the corresponding author. The data are not publicly available due to the research data governance policy of the institution that provided the ethical approval of the study.

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
