# Peer review of "Effects of Nutrition Counselling and Unconditional Cash Transfer on Child Growth and Family Food Security in Internally Displaced Person Camps in Somalia—A Quasi-Experimental Study"

_ijerph, 2022, doi:10.3390/ijerph192013441_

Round 1
Reviewer 1 Report
The article approaches still a current problem in different parts of the world. Children's malnutrition has not been solved, even if it has existed for a long time. It is an important factor that affects the population for a long time, considering that the young will become future adults, and some health problems may influence them further.
The authors are recommended to consider the following aspects:
· Use the journal template. The material uploaded is inadequate in respect of the presentation form.
· Reconsider the phrase construction, starting from the Abstract. Some of these seem to be unfinished.
· At the level of expression, there are considerable deficiencies. The material rewrite could be considered.
· The use of synonyms or rephrasing could be considered.
· Fig. 1 resolution needs improvement.
· Improvement of the reference section.
· Since the authors have concluded that the analyzed factors do not have a notable impact, they could include possible scenarios to increase the children's health in the material. All must cross-refer to the newest data presented in the defined area. It is important to consider the natural products locally available and how to maximize each family's income to obtain the desired finality.
The paper could be considered for publication after major changes. It has to be revised by the author(s) and resubmitted with suggested modifications specified in the reviewer’s comments.
Author Response
Dear,
Thank you so much for the review. Kindly find our comments for the revision.
Best Regards

Reviewer 2 Report
Comments to author(s):
Paper review “effects of nutrition counselling and unconditional cash transfer on child growth and family food security in internally dispersed person camps in Somalia - A quasi-experimental study”.
The research project is a study about nutrition and food security. It is very important in Somalia. The paper is well written and easy to understand.
Concerns raised include:
1. Abstract
Line 3, ‘To evaluate the effects of NC and NC combined with unconditional cash transfer on children’s growth and food security in Somalia.’ The sentence is not completed.
Lines 13-15, ‘A culturally tailored NC programme over longer period … might result in a significant reduction in malnutrition and food insecurity.’ You might need literature support for the statement.
2. Background
Line 8, it is better to use the full name for UNHCR.
Line 19, ‘Nutrition counselling part of strategies to reduce malnutrition in children.’ The sentence is not completed.
3. Methods
Study design
Ethics approval and informed consent should be included in the methods.
The study used two intervention groups and one control group. The control group did not receive intervention.
If an intervention is expecting to have positive impact on health, it is better to have all the participants receiving the intervention. The appropriate study design is stepped wedge. It is my suggestion for future research.
Sample size
design effect of 2.0 (ref), please include the reference.
4. Discussion
There were limitations in the study as you discussed in the manuscript. How did the caregivers spend the money is also an influence factor, as you mentioned that the quality of the training and supervision of CNVs is very important.
Please check your contexts ‘one possible explanation for our intervention not showing any improvement … in different seasons. The baseline survey was conducted during the dry season, and the follow up was conducted during the dry season.’
5. Other comments
Please read the paper carefully with regard to correct English.

Author Response
Dear,
Thank you so much for the review. Kindly find our revision for the comments.

Round 2
Reviewer 1 Report
The authors implemented most of the recommendations made.
There are still deficiencies at the level of English statements. The same words are repeated in the current or consequent phrase. Rephrase or use synonyms.
The paper could be considered for publication after minor changes. It has to be revised by the author(s) and resubmitted with suggested modifications specified in the reviewer’s comments.